# Interactions between stimulus and response types are more strongly represented in the entorhinal cortex than in its upstream regions in rats

**Eun-Hye Park[†], Jae-Rong Ahn[†], Inah Lee***

Department of Brain and Cognitive Sciences, Seoul National University, Shillim-dong, Korea

**Abstract** Previously we reported results which suggested that response types are critical in dissociating the lateral entorhinal cortex (LEC) from the medial entorhinal cortex (MEC) in a scene memory task (*Yoo and Lee, 2017*). Here, we investigated whether the perirhinal cortex (PER) and postrhinal cortex (POR), the upstream regions of the LEC and MEC, respectively, could be dissociated similarly. We conducted four tasks by combining different stimulus and response types. Our results suggest that the PER is important whenever object recognition is required and, together with prior findings, imply that PER-LEC networks are essential in goal-directed interactions with objects. The POR appears critical for recognizing visual scenes and may play key roles in scene-based navigation together with the MEC. The relative lack of functional dissociation between stimulus and response types at the PER-POR level suggests that actions conditioned on the recognition of external stimuli may be uniquely represented from the EC.
DOI: https://doi.org/10.7554/eLife.32657.001

**\*For correspondence:**
inahlee@snu.ac.kr

[†]These authors contributed equally to this work

**Competing interests:** The authors declare that no competing interests exist.

## Introduction

A prevailing theory posits that spatial information travels from the postrhinal cortex (POR) to the medial entorhinal cortex (MEC), and nonspatial information is transmitted from the perirhinal cortex (PER) to the lateral entorhinal cortex (LEC), before both types of information merge in the hippocampus (*Knierim et al., 2014*). An important caveat of this theory is that it does not explain explicitly how visual scene information is processed in these circuits, although it becomes increasingly clear that visual scenes may provide critical contextual information to the hippocampus (*Dombeck et al., 2010*; *Hassabis and Maguire, 2009*; *Kim et al., 2012*; *Maguire and Mullally, 2013*; *Prusky et al., 2004*; *Wirth et al., 2003*; *Zeidman et al., 2015*).

Recently, we reported that the LEC and MEC can be functionally dissociated in scene-memory tasks (*Yoo and Lee, 2017*). Interestingly, we found that sensory cues interact with task-specific response types. Specifically, we found that the LEC, but not the MEC, is critical when rats were required to manipulate a common object in different ways (by pushing it or digging sand) using visual scenes in the background, whereas the MEC, but not the LEC, was important when rats made spatial choices in the T-maze using the same visual scenes.

Building on our prior findings, we here sought to address a critical question, namely, whether such dissociation uniquely occurs at the level of the EC or is inherited from its upstream structures. We tested this by pharmacologically manipulating the PER and POR, the direct upstream regions of the LEC and MEC, respectively, and testing rats in the same scene-based testing paradigms applied in the Yoo and Lee study. Furthermore, as a comparison to the null results reported in the LEC and

MEC in object-based behavioral tasks in our previous study, we investigated whether the PER and POR play a role in such tasks (*Yoo and Lee, 2017*).

## Results

The following four different behavioral tasks, named based on stimulus and response types, were used in the current study: (i) scene-cued nonspatial response (SCN-NSR) task, (ii) scene-cued spatial response (SCN-SR) task, (iii) object-cued nonspatial response (OBJ-NSR) task, and (iv) object-cued spatial response (OBJ-SR) task. Rats (n = 23) trained to associate different types of stimuli (visual scenes and objects) with responses (spatial and nonspatial responses) were implanted bilaterally with cannulae in both the PER and POR (*Figure 1A*). Individual tip locations of cannulae were histologically verified (*Figure 1B*), and data from rats whose cannula-tip locations were misplaced from either area were discarded (n = 2).

### Similar contributions of the PER and POR to visual scene-based nonspatial responses

Rats (n = 7) were trained in the SCN-NSR task as described previously (*Yoo and Lee, 2017*) (*Figure 2A*). Injection of muscimol (MUS) into either the PER or POR (0.5 µL per site) resulted in significant performance deficits compared with injection of the same volume of artificial cerebrospinal fluid (ACSF) (*Figure 2B*; *Figure 2—source data 1*). A one-way repeated-measures ANOVA showed a significant effect of drug ($F_{(2,12)}$ = 11.363, p=0.001). A Bonferroni-Dunn post hoc test (corrected $\alpha$ = 0.017) revealed significant differences in performance between ACSF and PER-MUS groups (p<0.01) and between ACSF and POR-MUS groups (p<0.001), but not between PER-MUS and POR-MUS groups (p>0.1) (*Figure 2B*). These results suggest that, unlike in the LEC and MEC (*Yoo and Lee, 2017*), both the PER and POR may play some role in nonspatial behavioral choices made using visual background scenes.

### Both the PER and POR are involved in making visual scene-based spatial responses

To examine whether the type of response interacts with the scene stimulus, as has been observed between the LEC and MEC (*Yoo and Lee, 2017*), we tested a separate group of rats (n = 8) in the SCN-SR task (*Figure 2C*). We found that inactivating either the PER or POR yielded similar performance deficits compared with controls (*Figure 2D*; *Figure 2—source data 2*). A one-way repeated-measures ANOVA showed a significant effect of drug ($F_{(2,14)}$ = 12.745, p<0.001). Performance decreased significantly in both the PER-MUS (p<0.01) and POR-MUS (p<0.001) group, compared with the ACSF group. Performance was similar between PER-MUS and POR-MUS groups (p=0.133; Bonferroni-Dunn) (*Figure 2D*). These results indicate that both regions contribute to scene-based spatial choice behavior. Importantly, unlike the case for the LEC and MEC (*Yoo and Lee, 2017*), a functional interaction between scene and response type was not observed in the PER or POR.

### Greater contribution of the PER than the POR to object-based memory tasks

In our previous study (*Yoo and Lee, 2017*), we reported that the LEC and MEC were not involved in the OBJ-NSR task. To test the involvement of the PER and POR in the same task, we trained the same rats (n = 7) used in the SCN-NSR task in the OBJ-NSR task (*Figure 3A*). Inactivation of the PER caused a noticeable drop in performance compared with controls (*Figure 3B*), whereas similar deficits were not found in the POR-MUS group. A one-way repeated-measures ANOVA showed a significant effect of drug ($F_{(2,12)}$ = 15.548, p<0.001); a significant difference in performance was also found between ACSF and PER-MUS groups (p<0.001) and between PER-MUS and POR-MUS groups (p<0.001; Bonferroni-Dunn) (*Figure 3B*; *Figure 3—source data 1*). Interestingly, allowing rats to sample the object only visually in the same task recruited the PER similarly and, to some extent, the POR as well (see *Figure 3—figure supplement 1*, *Figure 3—figure supplement 1—source data 1* for additional details).

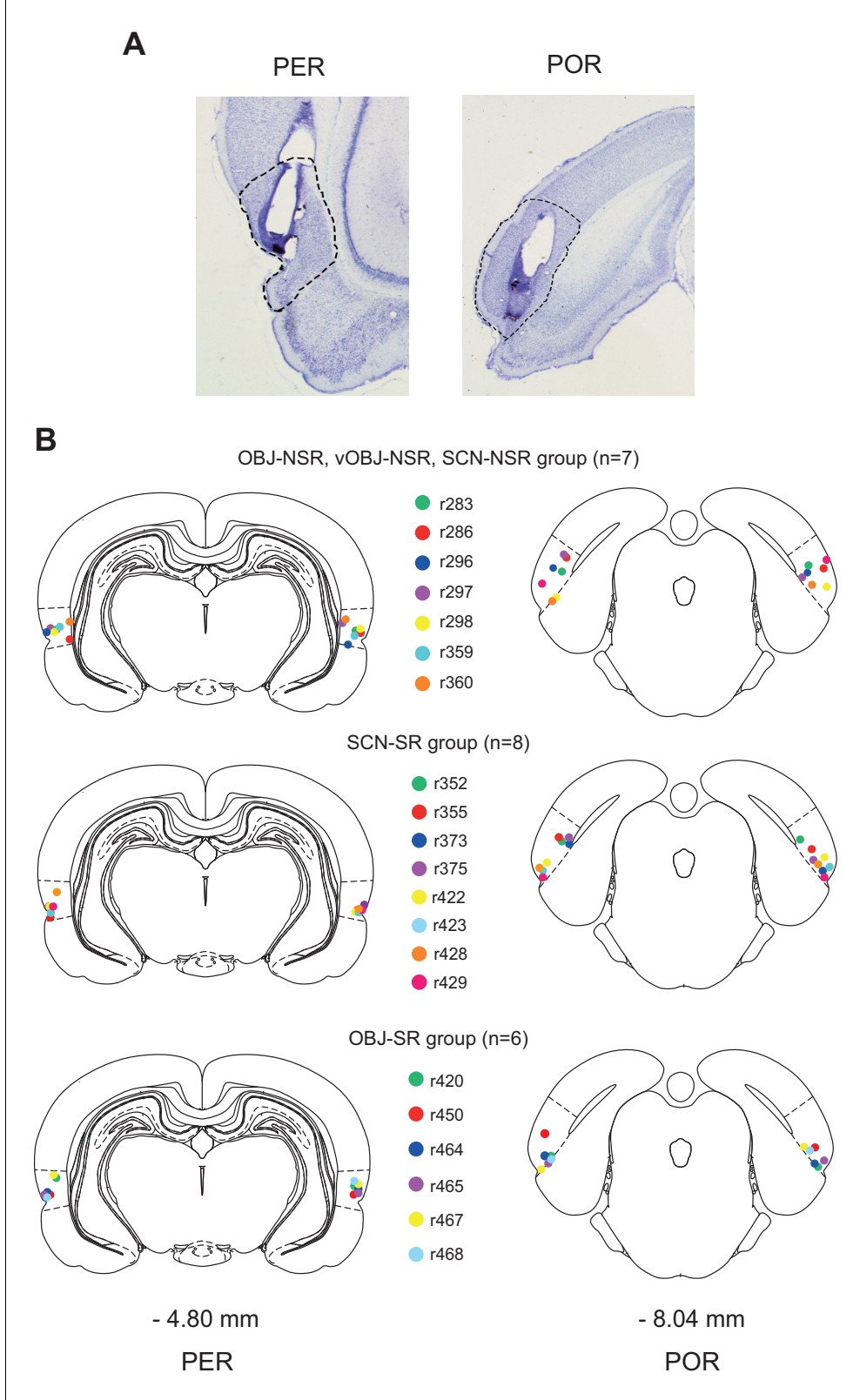

**Figure 1.** Cannula implantation in the PER and POR. (**A**) Cannula tracks in thionin-stained sections in the PER and POR. (**B**) Cannula-tip positions are indicated by dots (color-coded for rats) for the nonspatial response tasks (top), SCN-SR task (middle), and OBJ-SR task (bottom).

DOI: https://doi.org/10.7554/eLife.32657.002

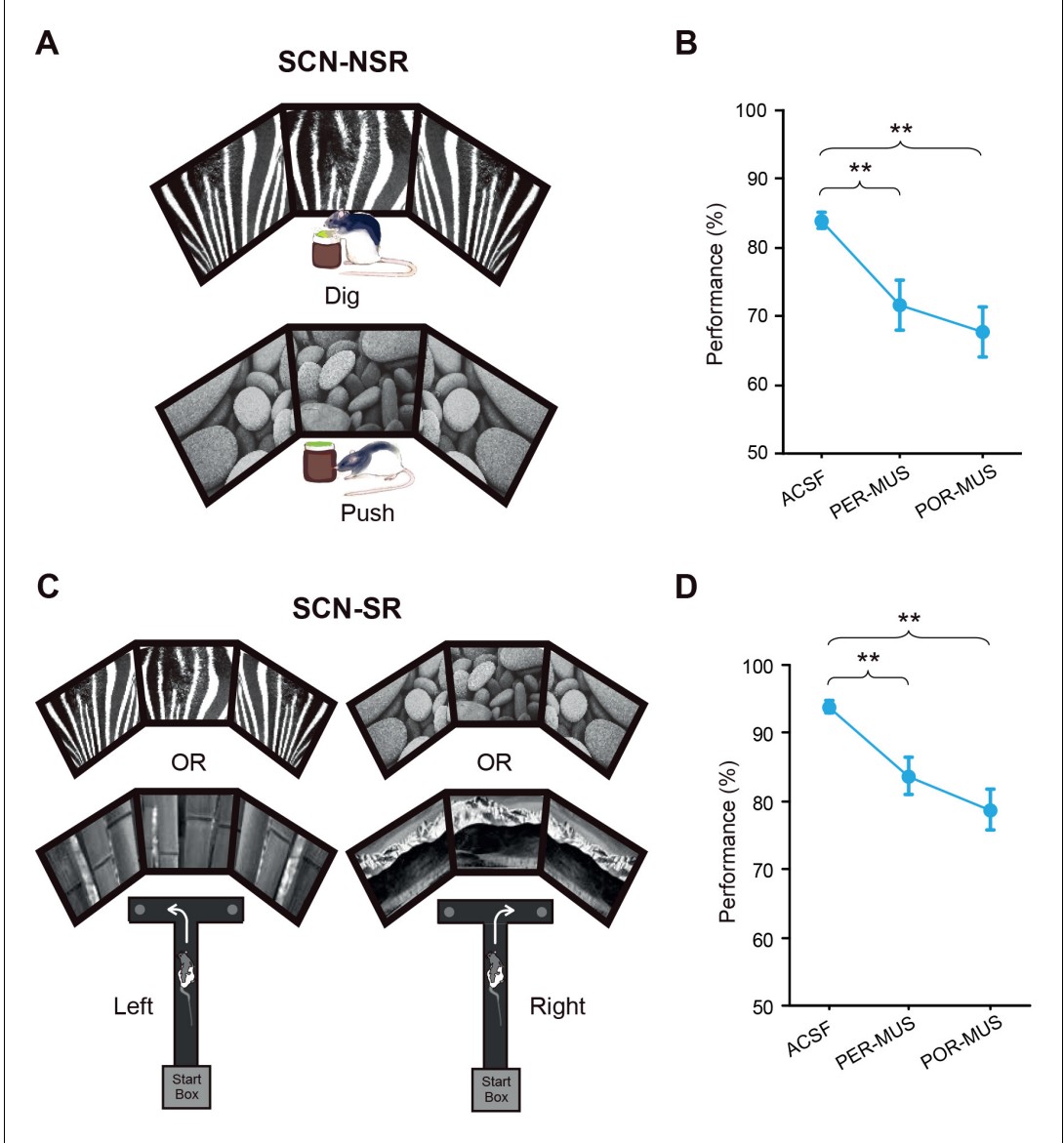

**Figure 2.** Scene-memory tasks. (**A**) Scene-cued nonspatial response task. Rats made a nonspatial response (push or dig) depending on the visual scene displayed on monitors. (**B**) Behavioral performance in the SCN-NSR task (Mean ± SEM). Both the PER- and POR-MUS conditions produced significant differences in performance compared to controls. (**C**) Scene-cued spatial response task. Rats made a spatial choice (left or right turn) depending on visual scenes. (**D**) Behavioral performance in the SCN-SR task (Mean ± SEM). Both the PER-MUS and POR-MUS conditions resulted in significantly impaired performance compared to control conditions. **p<0.01.

DOI: https://doi.org/10.7554/eLife.32657.003

The following source data is available for figure 2:

**Source data 1.** Performance in the SCN-NSR task.
DOI: https://doi.org/10.7554/eLife.32657.004
**Source data 2.** Performance in the SCN-SR task.
DOI: https://doi.org/10.7554/eLife.32657.005

We further tested whether the PER was important when spatial choices were required upon recognizing an object, because the PER may be engaged in a task as long as an object is used as a cue, regardless of the response type. For this purpose, we trained another group of rats (n = 6) in the OBJ-SR task, in which rats learned to make spatial choices using a toy object as a cue (**Figure 3C**). Rats exhibited severe performance deficits in the PER-MUS group, as was the case in the OBJ-NSR

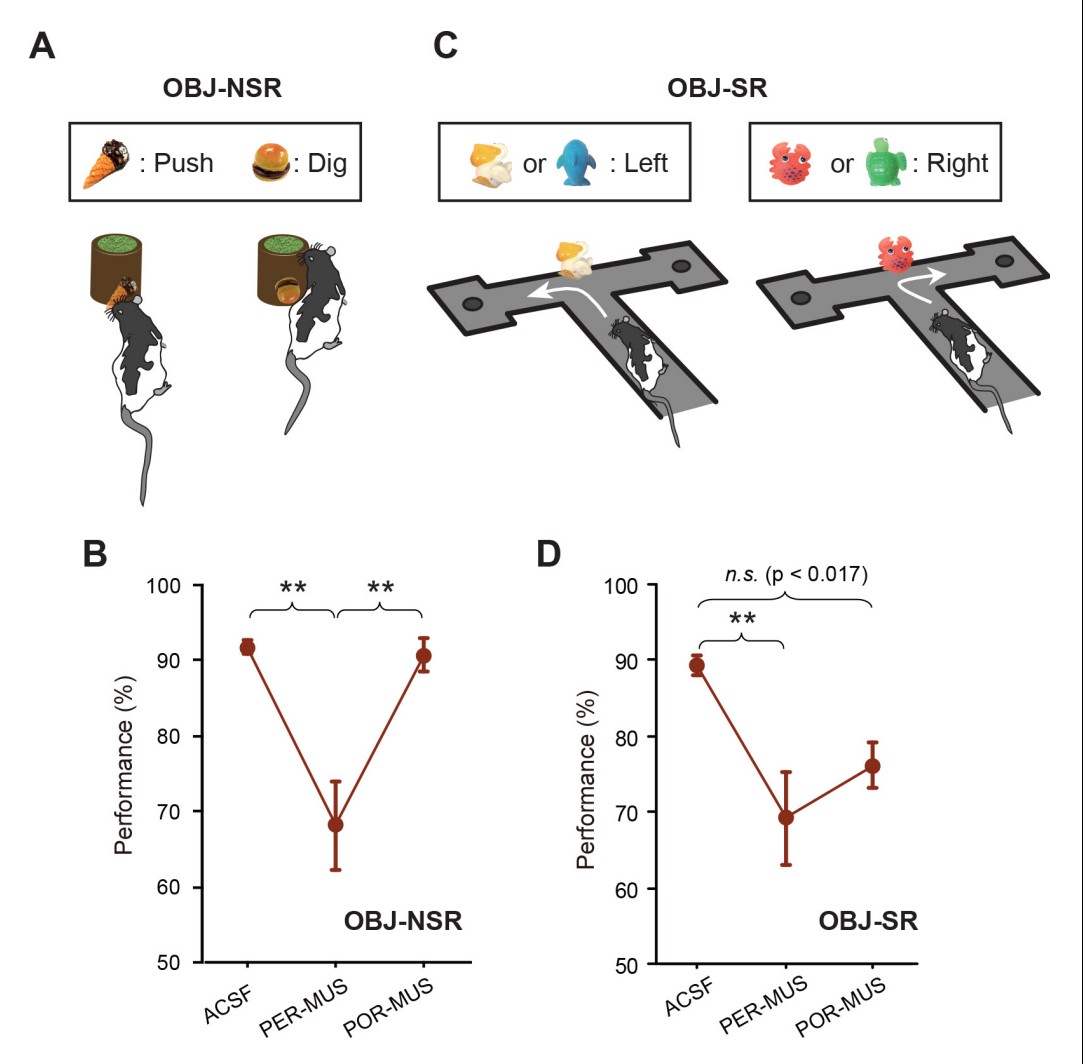

**Figure 3.** Object-memory tasks. (**A**) Rats made a nonspatial choice (push or dig) toward the sand-filled jar depending on the object cue attached to the jar. (**B**) Behavioral performance in the OBJ-NSR task (Mean ±SEM). The PER-MUS condition resulted in significant deficits in performance compared to the ACSF and POR-MUS conditions. (**C**) Object-cued spatial response task. Rats made a spatial choice depending on the toy object attached to the intersection wall. (**D**) Behavioral performance in the OBJ-SR task (Mean ± SEM). A significant difference in performance was found between the ACSF and PER-MUS conditions. **p<0.01.

DOI: https://doi.org/10.7554/eLife.32657.006

The following source data and figure supplements are available for figure 3:

**Source data 1.** Performance in the OBJ-NSR task.

DOI: https://doi.org/10.7554/eLife.32657.009

**Source data 2.** Performance in the OBJ-SR task.

DOI: https://doi.org/10.7554/eLife.32657.010

**Figure supplement 1.** Multimodal versus visual OBJ-NSR tasks.

DOI: https://doi.org/10.7554/eLife.32657.007

**Figure supplement 1—source data 1.** Performance in the visual OBJ-NSR task.

DOI: https://doi.org/10.7554/eLife.32657.008

task (*Figure 3D*; *Figure 3—source data 2*). Compared with the OBJ-NSR task, the POR-MUS group showed a trend toward mild deficits that failed to reach statistical significance. Specifically, a one-way repeated-measures ANOVA showed a significant effect of the drug ($F_{(2,10)}$ = 7.611, p<0.01) and a significant difference only between ACSF and PER-MUS groups (p<0.005), but not between ACSF

and POR-MUS groups (p=0.034, a value that did not reach the corrected α = 0.017 in Bonferroni-Dunn tests) (*Figure 3D*).

## Functional dissociations are less clear in the upstream cortical regions of the EC

To compare the current findings with functional dissociations previously observed in the MEC and LEC (*Yoo and Lee, 2017*), we plotted the relative performance deficits under MUS conditions (compared to ASCF conditions) in all tasks by subtracting the performance level under MUS from that in control conditions (*Figure 4A*).

In the NSR tasks (SCN-NSR and OBJ-NSR), the PER-MUS group showed large performance deficits in the OBJ-NSR task. The performance deficits in the SCN-NSR task were relatively smaller than those in the OBJ-NSR task. The POR-MUS group did not show any performance deficits in the OBJ-NSR task, but showed significant deficits in the SCN-NSR task (*Figure 4A*). A comparison of performance between OBJ-NSR and SCN-NSR tasks (two-way repeated measures ANOVA with task and region as within-subject factors) revealed a significant effect of region ($F_{(1,6)} = 30.295$, p<0.01), but not task ($F_{(1,6)} = 0.08$, p>0.5). The interaction between the two factors was significant ($F_{(1,6)} = 6.561$, p<0.05). A post hoc t-test (α = 0.025) showed a significantly larger deficit in the SCN-NSR than in the OBJ-NSR task in the POR (p<0.025), but not in the PER (p>0.1) (*Figure 4A*). Similar results were also obtained when only visual sampling was allowed in the OBJ-NSR task (*Figure 4—figure supplement 1A*).

In SR tasks (SCN-SR and OBJ-SR), the PER-MUS group showed larger deficits in the OBJ-SR task than in the SCN-SR task, whereas in the POR-MUS group, the results were similar between the two tasks (*Figure 4B*). There were no significant effects of region ($F_{(1,12)} = 0.470$, p=0.506) or task ($F_{(1,12)} = 0.872$, p=0.368), but the interaction between the two factors approached significance ($F_{(1,12)} = 4.338$, p=0.059).

Taken together, these findings strongly indicate that whether an object functions as a cue in a task may determine whether the PER plays an important role in the task, whereas the associated type of response may not be so critical. In contrast, the type of response (e.g., spatial navigational response) may play a greater role in recruiting the POR than the type of the cueing stimulus. An important finding in our study is that there is a condition in which the POR may not be necessary, namely the OBJ-NSR task (*Figure 4A*). One exception might be when a purely visual object is used as a cueing object (see below for further discussion). Most importantly, it is clear that the interaction between the stimulus attribute and response type between the PER and POR is not as strong as between the LEC and MEC (*Figure 4C*; *Figure 4—figure supplement 1B and C*).

## Discussion

In the current study, we sought to investigate functional differences between the PER and POR, the upstream regions of the LEC and MEC, respectively. Unlike in the LEC and MEC (*Yoo and Lee, 2017*), functional dissociations based on the response type (i.e., spatial vs. nonspatial) were less clear in the upstream areas in the scene-based tasks. Instead, the type of stimulus (i.e., object and scene) appears to critically determine the involvement of these areas in a task (*Figure 4*). Specifically, the PER was important in cases where an object stimulus should be recognized, but less so for scenes, regardless of the response type, whereas the POR was involved when either visual scene recognition or spatial navigation was necessary. Our findings suggest that the PER and POR may provide object and scene information to the LEC and MEC, respectively. In the EC, this information may be associated with task-demand–specific actions involving interaction with an object (LEC) or navigating in space (MEC). That is, the PER-LEC and POR-MEC networks may process 'what should I do in relation to this object?' and 'where should I go from here?', respectively (*Figure 4C*).

Anatomically, the PER receives multimodal sensory inputs (e.g., auditory, visual, olfactory, and somatosensory) from various sensory-perceptual cortices, whereas the POR receives major inputs largely from visual areas, including the retrosplenial cortex (*Agster and Burwell, 2009*; *Burwell and Amaral, 1998a*). In the current study, inactivating the PER indeed produced large deficits compared with controls whenever rats were required to recognize an object before making a spatial or nonspatial response. In contrast, inactivation of the POR resulted in reliable deficits in performance whenever scene was used as a conditional cue before making a spatial or nonspatial choice. Together

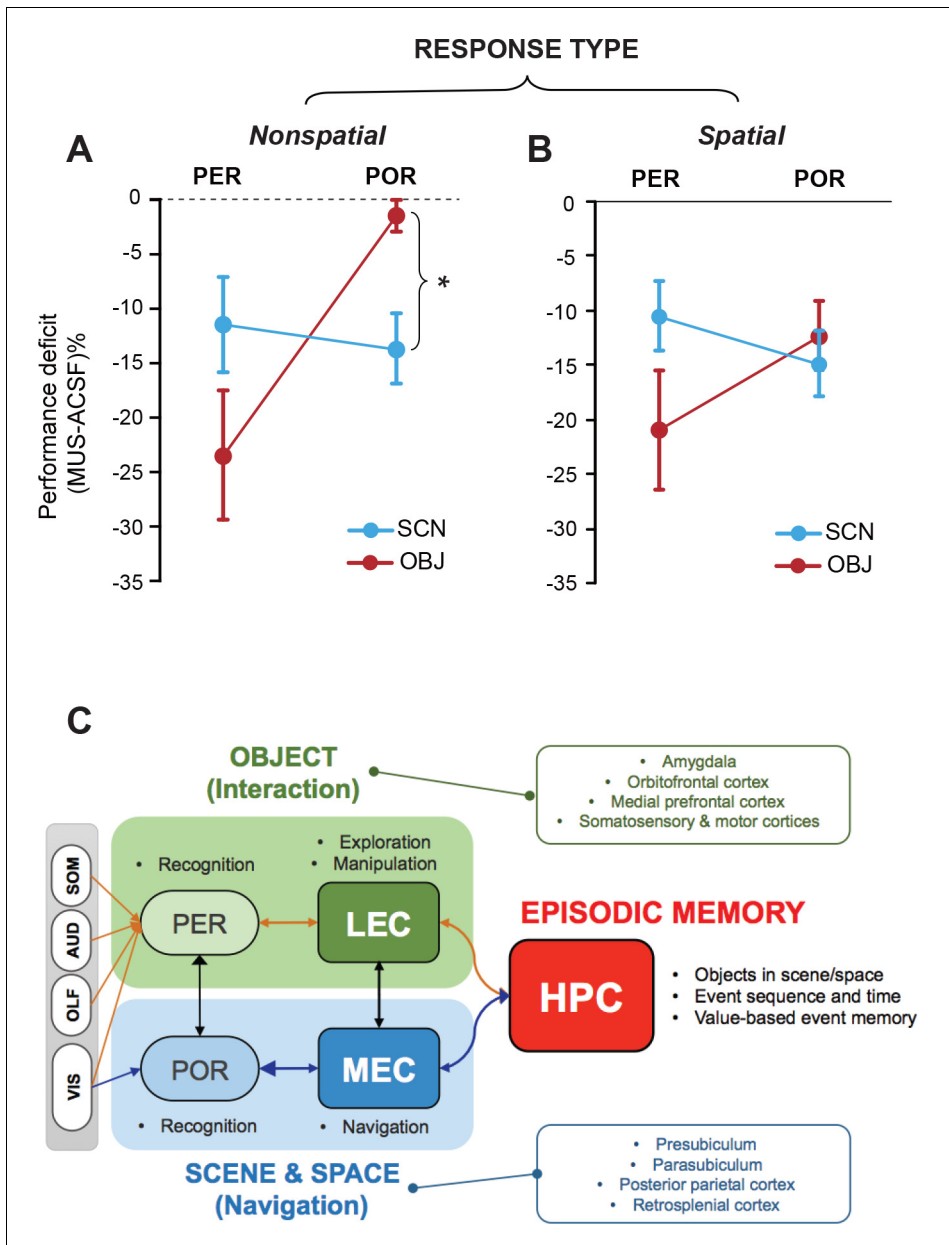

**Figure 4.** Lack of stimulus-response interaction in the PER and POR and a theoretical model. (**A**) Performance deficits (calculated by subtracting the ACSF-based performance from the MUS-based performance) in the nonspatial response tasks (Mean ± SEM). The POR-MUS condition resulted in significant deficits in performance when scenes were used as cues, but not when objects were used. The PER-MUS condition produced deficits in using scenes and objects for making nonspatial choices with bigger deficits with object cues. (**B**) Performance deficits in the spatial response tasks (Mean ± SEM). Both PER-MUS and POR-MUS conditioned produced similar levels of impairment observed in the nonspatial tasks (**A**), suggesting the lack of scene-response interaction at the PER and POR level. Furthermore, the more prominent roles of the PER, but not the POR, in the object-cued task was also observed in the spatial response tasks. *p<0.025. (**C**) A working model for information processing in the medial temporal lobe. Multimodal sensory inputs (VIS: visual, OLF: olfactory, AUD: auditory, SOM: somatosensory) are provided to the PER, and only visual inputs are fed to the POR. The PER and POR process these inputs to recognize objects and scenes, respectively. The LEC is involved in remembering choice responses associated with objects, whereas the MEC represents navigation-related variables using visual scene information from the POR. The LEC is reciprocally connected to the PER, hippocampus, insular cortex, and frontal areas (**Burwell and Amaral, 1998a**). Also, the LEC projects to the basal ganglia, medial prefrontal cortex, somatosensory cortex, and motor areas (**Swanson and Köhler, 1986**). The MEC has reciprocal connections with the POR, hippocampus,

*Figure 4 continued on next page*

*Figure 4 continued*

cingulate, and parietal cortex (*Burwell and Amaral, 1998b*). The MEC receives projections from the parasubiculum and postsubiculum (*Canto et al., 2008*). In this model, the PER-LEC networks are to interact with objects and the POR-MEC networks process information to navigate in space. In the hippocampus, the neural representations from these two channels are temporally structured with relative values in a goal-directed manner to generate rich episodic memories.

DOI: https://doi.org/10.7554/eLife.32657.011

The following figure supplement is available for figure 4:

**Figure supplement 1.** Comparisons of the PER-POR networks with the LEC-MEC networks.

DOI: https://doi.org/10.7554/eLife.32657.012

with prior findings (*Epstein and Kanwisher, 1998*; *Murray and Richmond, 2001*), our results suggest that the PER and POR may be specialized in recognizing (and perhaps perceiving) objects (*Ahn and Lee, 2017*; *McTighe et al., 2010*) and scenes, respectively.

We previously reported a strong double dissociation in the LEC and MEC between scene-based tasks that required nonspatial responses (i.e., object manipulation) and spatial responses (i.e., navigational turns) (*Figure 4—figure supplement 1B*) (*Yoo and Lee, 2017*). Testing rats in the same tasks with the PER or POR inactivated did not result in such strong dissociations in the current study (*Figure 4—figure supplement 1C*), suggesting that scene-response interactions may uniquely occur in the EC. However, when the stimulus category was extended to include objects, some interactions between stimulus and response types became apparent in the POR, but not necessarily in the PER. Specifically, a comparison of performance deficits between scene- and object-based tasks with the PER inactivated showed similarly larger deficits in object-based tasks (OBJ-NSR and OBJ-SR tasks) than in scene-based tasks (SCN-NSR and SCN-SR tasks), regardless of the response type. However, the navigational task demand in both OBJ-SR and SCN-SR tasks required the POR, irrespective of the type of stimulus, whereas the same region was only necessary when a visual scene (SCN-NSR task), but not an object (OBJ-NSR task), was used as a cue in nonspatial tasks. These results suggest that both scene recognition and spatial navigation constitute important computational components of the POR network (*Figure 4C*).

It is unclear why the navigational demand affected the POR regardless of the stimulus type, but had a lesser effect in the PER (*Figure 4B*). One possibility is that the POR indeed processes navigation-related signals in association with objects and scenes, based on the reciprocal connections with the MEC (*Figure 4C*). Because strong neural correlates for spatial navigation (e.g., grid cells and border cells) have been reported in the MEC (*Hafting et al., 2005*; *Sargolini et al., 2006*; *Solstad et al., 2008*), the MEC may bias the POR network toward processing incoming sensory inputs (e.g., visual object or scene information) in relation to the spatial navigation components of the task. Such a strong bias may not be exerted by the LEC to the PER, because object recognition may lead to many different types of responses in natural settings, as opposed to the case of scene recognition, which is normally associated with spatial navigation. Another possibility is that top-down influences from other navigation-related regions (e.g., retrosplenial cortex) might be stronger in the POR-MEC networks than in the PER-LEC networks. Considering the anatomical finding that the POR-to-PER connection is stronger than the PER-to-POR connection (*Burwell and Amaral, 1998b*; *Furtak et al., 2007*; *Kealy and Commins, 2011*), it is also possible that rats with inactivation in the POR might send disturbing signals to the PER, preventing normal object recognition in the OBJ-SR task (*Figure 3D*). Nonetheless, our study clearly demonstrates that the POR is not necessary if neither spatial navigation nor visual scene information processing is required.

Visual scenes used in our tasks may be considered as a context, and the PER-POR networks may play significant roles in contextual object recognition (*Norman and Eacott, 2005*; *Heimer-McGinn et al., 2017*). In natural situations, an animal never experiences an object completely detached from its background, and a visual context also always involves objects in it (*Aminoff et al., 2013*). Therefore, PER and POR networks may function in harmony in natural settings, rather than in isolation from each other. For example, an object is recognized efficiently when it appears against a contextually plausible background (*Aminoff et al., 2013*). Disrupting the POR network may affect such contextual object recognition (*Furtak et al., 2012*) in our SCN-NSR task because the 'meaning' (e.g., push or dig) of the object (i.e., jar) should be disambiguated using visual scenes in the

background in that task. Likewise, inactivating the PER as in our study may make it harder for the animal to focus on the target object against its background context. These functional relationships between the PER and POR may underlie the similar deficits between PER-MUS and POR-MUS groups in the SCN-NSR task. Inactivation of the PER may also have some disruptive effects on the POR's functions in scene information processing through PER-to-POR connections, leading to mild performance deficits (but still >80% correct) in the SCN-SR task in our study. Another possibility is that scene recognition in the POR may require some contribution of the PER, because a scene is normally composed of individual objects (e.g., individual pebble stones in the pebbles scene in our task). On a similar note, inactivation of the POR may make it difficult for an animal to focus on the cueing object in the OBJ-SR task, because contextual disruption, but not facilitation, of object recognition might occur.

In the view of traditional theory, spatial and nonspatial information is processed via separate streams in the medial temporal lobe (*Knierim et al., 2014*). According to this model, the physical attribute (i.e., spatial or nonspatial) of a stimulus determines the processing stream (i.e., PER-LEC or POR-MEC) to which the information is channeled. However, the current results, together with our previous findings, strongly suggest that such a view may be too simplistic. Our current working model is that the PER-LEC and POR-MEC networks are mainly concerned with object manipulation (or exploration) and spatial navigation (or contextual memory), respectively (*Figure 4C*). Our model posits that recognition of perceptual stimuli mainly occurs at the level of the PER (for objects) and POR (for scenes), and their task-related actions are associatively represented in the EC networks. The hippocampus, which receives these inputs together, may encode and retrieve various event memories in sequences as an animal navigates through a space to achieve goals.

## Materials and methods

### Subjects
Twenty-three male Long-Evans rats (8 weeks old) were obtained and housed individually in Plexiglas cages in a temperature- and humidity-controlled animal colony. Rats were maintained on a 12 hr light/dark cycle (lights on at 8:00 am), and experiments were carried out in the light phase of the cycle. Rats were food-restricted to maintain 80% of their free-feeding weight, but allowed access to water freely. All protocols for animal care and surgery adhered to the guidelines of the Institutional animal care and Use Committee of the Seoul National University (SNU-120925-1-7).

### Behavioral paradigm
Detailed procedures for the SCN-NSR, SCN-SR, and OBJ-NSR tasks can be found in the previous study (*Yoo and Lee, 2017*). The OBJ-SR task was performed in the same T-shaped linear track used in the SCN-SR task. One of the four 3-dimensional toy objects (dolphin, turtle, crab and duck) were affixed upright via a magnet at the intersection of the track, and the rat had to visit the arm associated with the object. The object stimuli used in the OBJ-SR task were somewhat bigger (6–7 cm in width and 7–8 cm in height) than those in the OBJ-NSR (3 cm in width and 3–7 cm in height) task to ensure that the rats properly sampled the objects before making a choice in the intersection.

### Bilateral cannulae implantation in the PER and POR
Four small burr holes were drilled and cannulae were implanted in the PER and POR bilaterally. The following coordinates were used for the first three animals: (1) PER: AP - 4.8 mm, ML ± 6.8 mm, DV - 4.6 mm from dura; (2) POR: AP – 8 mm, ML ± 6.5 mm, DV - 1.5 mm from dura in 3 animals. In order to minimize the damage in the temporal muscles, the coordinates were later revised for the rest of the animals: (1) PER: AP - 4.8 mm, ML ±5 mm, DV – 6 mm from dura with the tip angled at 15° laterally, and (2) POR: AP – 8 to 8.2 mm, ML ± 5 to 5.3 mm, DV - 3.4 to 4 mm from dura with the tip angled at 15° laterally.

## Acknowledgements

This work was supported by the National Research Foundation of Korea (2015M3C7A1031969, 2016R1A2B4008692, 2017M3C7A1029661) and the BK21 + program (5286–2014100). We thank Heung-Yeol Lim for his assistance in behavioral testing.

## Additional information

### Funding

| Funder | Grant reference number | Author |
| --- | --- | --- |
| National Research Foundation of Korea | 2015M3C7A1031969 | Inah Lee |
| National Research Foundation of Korea | 2016R1A2B4008692 | Inah Lee |
| National Research Foundation of Korea | 2017M3C7A1029661 | Inah Lee |
| National Research Foundation of Korea | 5286-2014100 (BK21+ program) | Inah Lee |

The funders had no role in study design, data collection and interpretation, or the decision to submit the work for publication.

### Author contributions

Eun-Hye Park, Conceptualization, Data curation, Software, Formal analysis, Visualization, Methodology, Writing—original draft; Jae-Rong Ahn, Data curation, Software, Formal analysis, Visualization, Methodology, Writing—original draft, Writing—review and editing; Inah Lee, Conceptualization, Resources, Supervision, Funding acquisition, Validation, Investigation, Visualization, Methodology, Writing—original draft, Project administration, Writing—review and editing

### Author ORCIDs

Inah Lee http://orcid.org/0000-0003-3760-4257

### Ethics

Animal experimentation: All of the animals were handled according to approved institutional animal care and use committee (IACUC) protocols of the Seoul National University (SNU-120925-1-7). All surgery was performed under isoflurane anesthesia, and every effort was made to minimize suffering.

### Decision letter and Author response

Decision letter https://doi.org/10.7554/eLife.32657.016
Author response https://doi.org/10.7554/eLife.32657.017

## Additional files

### Supplementary files

• Transparent reporting form
DOI: https://doi.org/10.7554/eLife.32657.013

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
