## [Decision Letter]

Thank you for submitting your article "Interactions between stimulus and response types are uniquely represented in the entorhinal cortex, but not in its upstream cortical regions" for consideration by *eLife*. Your article has been reviewed by two peer reviewers, and the evaluation has been overseen by a Reviewing Editor and Sabine Kastner as the Senior Editor. The following individual involved in review of your submission has agreed to reveal his identity: Alex Easton (Reviewer #1).

The reviewers have discussed the reviews with one another and the Reviewing Editor has drafted this decision to help you prepare a revised submission.

Summary:

The current study is a follow-up to work done recently, published in *eLife*, in which these authors showed a dissociation in the lateral EC (LEC) from the medial EC (MEC) in a scene memory task based on response type. Here the authors test whether a similar dichotomy in processing information exists at the two regions directly upstream from these areas, the perirhinal and postrhinal regions. Each region was inactivated across several tasks using either object or response type. The authors found that the clean dissociation present at EC was not fully reproduced, and that instead there was substantial overlap in the involvement of the two areas across task requirements.

Essential revisions:

Both reviewers were overall positive and felt that the study was a logical and appropriate follow-up to the prior work, and that the experiments were well designed and executed. That the data did not cleanly match predictions nor detract from enthusiasm for the experiment and desire to see it published. However in discussions, it was felt that the authors could be clearer in describing some of the results (see reviews). In addition, and probably more critically, it was felt that the authors need to go beyond their current conclusions and provide a model or proposal of some sort to explain the unexpected results. As one reviewer put it, they should "explain how their results cannot be explained by existing models and therefore which part(s) of the existing models need changing….. what they think might be going on (is it feedback from ERC that means these regions are harder to dissociate further upstream? Is it some anatomical crossover at the PER/POR level that differentiates the effects from the clearer dissociations in ERC? Is it that only in ERC does processing require these components to be separated and before that there is just less going on? Is it a difficulty effect between scenes and objects?)". Basically propose some testable ideas for why the segregation of information is present at one level but not another. These were the essential revisions I think.

Reviewer #1:

This is a concise, but important, follow up study to the authors' 2017 paper. Using spatial and non-spatial tasks in which the instruction cues are either objects (visual or multisensory) or scenes they investigate the role of Perirhinal and Postrhinal cortices through pharmacological inactivation of these regions.

The authors find that inactivation of the perirhinal cortex impairs performance on object cued tasks more than scene cued tasks (for both spatial and non spatial tasks). Postrhinal lesions appear to have a similar effect on both object and scene cued tasks, except where the object is multisensory in nature, in which case for non spatial tasks there is little sign of impairment.

Whilst the paper is clear and the findings of great interest, there is much the authors leave unsaid which is important. They broadly discuss that existing theory needs to be updated – but it seems to me that the authors are in a position with their data now to make suggestions as to what that updated theory might look like. It is important that the authors interpret this data in terms of potential mechanistic explanations. In particular, at the moment they talk in general about POR being critical for recognising visual stimuli and the PER for objects in particular. However, it seems to me this might not reflect the full dissociation the authors suggest. It is possible that both PER and POR could have the same function, but with POR having a more significant effect, meaning that scenes and objects are impaired following POR lesions, but the lesser effects of PER lesions would impair objects but not scenes, where additional information might be able to be used to solve the task. There is no clear double dissociation here to be confident about the independence of these two regions.

In that theme, Figure 4 is extremely helpful, but highlights that scene guided learning is never more impacted by a lesion than object guided learning. This might provide support for the idea that scene guided learning is just easier, and that the apparent dissociation between POR and PER merely reflects this.

Reviewer #2:

The submitted manuscript fits the format of Research Advance papers published in *eLife* in the sense that it builds upon a Research Article recently published by the same authors in 2017, by using an experimental design directly inspired by the original data. The published paper was about the contribution of lateral and medial entorhinal cortex (LEC and MEC respectively) to spatial and non-spatial response in the context of a scene discrimination. In the present manuscript, the authors ask whether the cortical structures directly upstream LEC and MEC, i.e. perirhinal and postrhinal cortex (PER and POR, respectively) show the same dissociation as that observed in LEC and MEC for the processing of scene-based type of response.

To do so, they use a within-subject design in which rats received local intra-cerebral injections of muscimol in either PER of POR during specific trials of several tasks differing in either the type of stimuli (spatial scene or object) or the type of response required (spatial, i.e. turn left or right, or non-spatial, i.e. push or dig). The results do not reveal a dissociation between PER and POR inactivations with regard to the type of response in the spatial scene discriminations. Both inactivations induce a deficit no matter the response required, contrary to MEC and LEC inactivations. This is the main result shown in Figure 2 and Figure 4—figure supplement 1. In addition, the authors show a complex pattern of results (with stronger effects of PER than of POR inactivations) with regard to object-based discrimination but here again no clear effect of the type of response.

The experiments are very well done. The results are interesting but the way they are summarized in Figure 4 and analyzed in complex two-way and three-way ANOVAs (last section of results) is very complicated and therefore distract the reader from the main message (perfectly understandable from Figure 2 and very nicely summarized in Figure 4—figure supplement 1). That PER and POR inactivations may have different effects on object discrimination in combination with type of response (Figure 3 vs. 3E) is also interesting and may be worth being reported. However, the introduction of the condition "vOBJ-NSR" here makes interpretation of the whole pattern of result even more complicated. If the authors want to carry a clear take-home message with significant impact, I suggest they simplify their presentation as much as possible.

In addition, some of their interpretations of the results are weird. One such example is the section on the results of object-based spatial choice where they say that "A significant difference was found only between ACSF and PER-MUS groups (p < 0.005), and not between ACSF and POR". Not only the statistics they report in this paragraph seem to show that POR are indeed impaired (and do not show a difference with PER), but just looking at the performance points in Figure 3 together with the height of the SEM seems to indicate the existence of an impairment as great in POR than in PER. In contrast it is clear that there is no impairment in PER rats for the same object discrimination if a non-spatial response if required (Figure 3) while POR rats are clearly impaired. We thus have a dissociation here, though it is not the one that was expected.

Thus overall the paper brings interesting data that complement nicely those of the original paper, but my impression is that the important points are buried in such a way the main message is obscured. What we need here is a model, even sketchy, of how all this circuitry works and a possible interpretation of what makes that the type of response required from the rat is so important in triggering such complex effects in conjunction with the type of stimuli.

---

## [Author Response]

Reviewer #1:[…]Whilst the paper is clear and the findings of great interest, there is much the authors leave unsaid which is important. They broadly discuss that existing theory needs to be updated – but it seems to me that the authors are in a position with their data now to make suggestions as to what that updated theory might look like. It is important that the authors interpret this data in terms of potential mechanistic explanations. In particular, at the moment they talk in general about POR being critical for recognising visual stimuli and the PER for objects in particular. However, it seems to me this might not reflect the full dissociation the authors suggest. It is possible that both PER and POR could have the same function, but with POR having a more significant effect, meaning that scenes and objects are impaired following POR lesions, but the lesser effects of PER lesions would impair objects but not scenes, where additional information might be able to be used to solve the task. There is no clear double dissociation here to be confident about the independence of these two regions.

We thank the reviewer for this suggestion. In our revised manuscript, we have included our working model to explain the results of the current study and the Yoo and Lee study. Specifically, we have added a model diagram in Figure 4 and, using this model, provided a more detailed discussion of our results. The Discussion section has been rewritten to the model-based accounts of the data and literature. Examples of paragraphs are as follows:

“In the current study, we sought to investigate functional differences between the PER and POR, the upstream regions of the LEC and MEC, respectively. […] That is, the PER-LEC and POR-MEC networks may process “what should I do in relation to this object?” and “where should I go from here?”, respectively (Figure 4).”

“In the view of traditional theory, spatial and nonspatial information is processed via separate streams in the medial temporal lobe (Knierim et al., 2014). […] The hippocampus, which receives these inputs together, may encode and retrieve various event memories in sequences as an animal navigates through a space to achieve goals.”

In that theme, Figure 4 is extremely helpful, but highlights that scene guided learning is never more impacted by a lesion than object guided learning. This might provide support for the idea that scene guided learning is just easier, and that the apparent dissociation between POR and PER merely reflects this.

We agree with the reviewer that there might be some differences in task difficulty among the behavioral tasks in our current and prior studies. One thing we haven’t observed, however, is a systematic relationship between a certain stimulus type (e.g., scene) and performance level. For example, based on the control’s performance level (ACSF), the easiest task seems to be the SCN-SR task (ACSF: ~95% correct, Figure 2). However, in the SCN-NSR task, the control’s performance was approximately 10% lower than that in the SCN-SR task even though the same scene stimuli were used (Figure 2). When objects were used as cues in the OBJ-NSR and OBJ-SR tasks, the control’s performance stayed at ~90% correct level, which was a bit higher than in the SCN-SR task, but lower than in the SCN-SR task. Moreover, performance in the SCN-NSR task of the POR-MUS group was significantly impaired, whereas the same group’s performance was not affected in the OBJ-NSR task. This pattern of results was reversed in the PER-MUS group (Figure 4). Based on these results, we believe that task difficulty alone may not be able to explain our results.

Reviewer #2:[…]The experiments are very well done. The results are interesting but the way they are summarized in Figure 4 and analyzed in complex two-way and three-way ANOVAs (last section of results) is very complicated and therefore distract the reader from the main message (perfectly understandable from Figure 2 and very nicely summarized in Figure 4—figure supplement 1). That PER and POR inactivations may have different effects on object discrimination in combination with type of response (Figure 3 vs. 3E) is also interesting and may be worth being reported. However, the introduction of the condition "vOBJ-NSR" here makes interpretation of the whole pattern of result even more complicated. If the authors want to carry a clear take-home message with significant impact, I suggest they simplify their presentation as much as possible.

We thank the reviewer for the constructive comments. This was one of our concerns as well as we put together our original manuscript. Following the reviewer’s comments, we decided to remove the vOBJ-NSR data from the main text in the revised manuscript (moved to our revised Figure 3—figure supplement 1 and Figure 4—figure supplement 1). We think this makes sense also because rats might rarely experience an object using visual modality alone in natural settings, and the data from the vOBJ-NSR task may be better suited for supplementary information.

In addition, some of their interpretations of the results are weird. One such example is the section on the results of object-based spatial choice where they say that "A significant difference was found only between ACSF and PER-MUS groups (p < 0.005), and not between ACSF and POR". Not only the statistics they report in this paragraph seem to show that POR are indeed impaired (and do not show a difference with PER), but just looking at the performance points in Figure 3 together with the height of the SEM seems to indicate the existence of an impairment as great in POR than in PER. In contrast it is clear that there is no impairment in PER rats for the same object discrimination if a non-spatial response if required (Figure 3) while POR rats are clearly impaired. We thus have a dissociation here, though it is not the one that was expected.

This is a good point. The performance of the POR-MUS group in the OBJ-SR indeed look lower than that of controls. In this study, we used the Bonferroni/Dunn post-hoc test for all post-hoc comparisons. The significance criterion (α) for the Dunn’s post-hoc test was 0.0167, and only the difference between the ACSF and PER-MUS (p = 0.0037), but not between the ACSF and POR-MUS (p = 0.034) met this criterion. Other conservative tests such as the Scheffe and Tukey/Kramer test also resulted in nonsignificant differences between the ACSF and PER-MUS groups. However, if we use more liberal post-hoc tests such as Fisher’s PLSD and Student Newman-Keuls, the difference became significant. In our prior study (Yoo and Lee, 2017), we used the Bonferroni-Dunn test to interpret the results more conservatively and to minimize the risk of making Type I error. Since the current study examined the same phenomena observed in the EC in the PER and POR, we would like to keep the same post-hoc tests in the current study. In the revised manuscript, we added more information in the graph (Figure 3) to explicitly show the p-value.

Nonetheless, although the Dunn’s test called the results insignificant, we agree with the reviewer that this is the case where one can see at least the trend of impairment. In the revised manuscript, we reflected this as we develop a working model to explain the results from the current and previous studies. Specifically, our revised manuscript states that the POR could be involved when scene recognition or spatial navigation (separately or together) is required in a task to include the possibility that POR inactivation actually could lead to significant performance deficits in the OBJ-SR task if sampling size increases. We now incorporate this in our newly presented model (see Figure 4 and its associated texts in Discussion) and explicitly states this point in the summary part of the discussion as follows:

“… Unlike in the LEC and MEC (Yoo and Lee, 2017), functional dissociations based on the response type (i.e., spatial vs. nonspatial) were less clear in the upstream areas in the scene-based tasks. Instead, the type of stimulus (i.e., object and scene) appears to critically determine the involvement of these areas in a task (Figure 4). Specifically, the PER was important in cases where an object stimulus should be recognized, but less so for scenes, regardless of the response type, whereas the POR was involved when either visual scene recognition or spatial navigation was necessary…”

The almost complete lack of impairment in the POR-MUS condition in the OBJ-NSR task indeed served as one of the clear results that helped us to build a hypothetical working model in our revised manuscript. That is, as stated above if neither visual scene nor spatial navigation is required, the POR doesn’t seem to be interested in the task (e.g., object-based digging or pushing an object). This point has also been added to our Discussion as we introduce our model and explain it in the revised manuscript.

Thus overall the paper brings interesting data that complement nicely those of the original paper, but my impression is that the important points are buried in such a way the main message is obscured. What we need here is a model, even sketchy, of how all this circuitry works and a possible interpretation of what makes that the type of response required from the rat is so important in triggering such complex effects in conjunction with the type of stimuli.

We thank the reviewer for the suggestion. Please see our responses to reviewer #1.